**DOI: 10.1038/ncomms16073**　　**OPEN**

# Enhanced anti-tumour immunity requires the interplay between resident and circulating memory CD8+ T cells

Michel Enamorado[1], Salvador Iborra[1], Elena Priego[1,2], Francisco J. Cueto[1,2], Juan A. Quintana[1], Sarai Martínez-Cano[1], Ernesto Mejías-Pérez[3], Mariano Esteban[3], Ignacio Melero[4,5], Andrés Hidalgo[1,6] & David Sancho[1]

The goal of successful anti-tumoural immunity is the development of long-term protective immunity to prevent relapse. Infiltration of tumours with CD8+ T cells with a resident memory (Trm) phenotype correlates with improved survival. However, the interplay of circulating CD8+ T cells and Trm cells remains poorly explored in tumour immunity. Using different vaccination strategies that fine-tune the generation of Trm cells or circulating memory T cells, here we show that, while both subsets are sufficient for anti-tumour immunity, the presence of Trm cells improves anti-tumour efficacy. Transferred central memory T cells (Tcm) generate Trm cells following viral infection or tumour challenge. Anti-PD-1 treatment promotes infiltration of transferred Tcm cells within tumours, improving anti-tumour immunity. Moreover, Batf3-dependent dendritic cells are essential for reactivation of circulating memory anti-tumour response. Our findings show the plasticity, collaboration and requirements for reactivation of memory CD8+ T cells subsets needed for optimal tumour vaccination and immunotherapy.

[1] Centro Nacional de Investigaciones Cardiovasculares Carlos III (CNIC), Melchor Fernández Almagro, 3, Madrid 28029, Spain. [2] Universidad Autónoma de Madrid, Arzobispo Morcillo 4, Madrid 28029, Spain. [3] Department of Molecular and Cellular Biology, Centro Nacional de Biotecnología, Consejo Superior de Investigaciones Científicas (CNB-CSIC), Darwin 3, Madrid 28049, Spain. [4] Division of Immunology and Immunotherapy, Center for Applied Medical Research (CIMA), Centro de Investigación Biomédica en Red de Cáncer (CIBERONC), 31008 Pamplona, Spain. [5] University Clinic, University of Navarra and Instituto de Investigación Sanitaria de Navarra (IdISNA), Pío XII, 55, 31008 Pamplona, Spain. [6] Institute for Cardiovascular Prevention (IPEK), Ludwig-Maximilians-Universität, Pettenkoferstrasse 9, 80336 Munich, Germany. Correspondence and requests for materials should be addressed to D.S. (email: dsancho@cnic.es).

Generation of optimal cancer immunotherapy involves induction of effective memory against the primary tumour able to prevent relapse metastases and recurrence. Circulating memory cells patrol the blood and include central memory T (Tcm) cells that retain the capacity to enter lymph nodes (LNs). Conversely, tissue-resident memory T (Trm) cells are confined to parenchymal non-lymphoid tissues[1–7]. Trm are characterized by stable surface expression of CD69 and an enhanced effector ability that functionally provides a tissue-wide alert state against local reinfection[6–11].

In mice, cutaneous infection with recombinant vaccinia virus (rVACV) generates circulating memory CD8[+] T cells and skin Trm cells, whereas i.p. infection does not generate skin Trm cells[12]. Infected parabiotic mice with skin Trm cells are more resistant to a rechallenge dermal infection than their circulation-sharing partners lacking Trm cells[12]. Optimal generation of Trm cells requires Batf3-dependent dendritic cells (DCs) during priming following VACV infection[13]. Batf3[−/−] mice show impaired immunity against syngeneic fibrosarcomas with marked intrinsic immunogenicity[14]. Tumour infiltration by CD103[+] Batf3-dependent DCs correlates with tumour regression[15] and favours T-cell infiltration in mouse models of melanoma[16]. CD103[+] DCs mediate antigen capture within the tumour and cross-prime tumour-specific CD8[+] T cells, whose therapeutic effects can be amplified by immunostimulatory antibodies[17,18].

The interplay between circulating CD8[+] T cells and Trm cells in anti-tumour immunity is largely unexplored. Previous studies in human cancer show that the infiltration of tumours by T cells with a Trm cell-like phenotype correlates with improved overall survival in early stage non-small-cell lung carcinoma, pulmonary squamous cell carcinoma and high-grade serous epithelial ovarian cancer[19–21]. In addition, recent results suggest that vaccination routes that promote generation of Trm cells could be more effective for anti-tumour response[22,23]. These findings prompted us to analyse the relative contribution and plasticity of circulating memory CD8[+] T cells and Trm cells in a model of anti-tumour vaccination.

In the present study, we demonstrate that circulating CD8[+] T cells and Trm cells cooperate in anti-tumour immunity. The circulating memory compartment retains enough degree of plasticity to become cells with a Trm phenotype within the grafted tumour and reside in the skin after tumour elimination. Immunotherapy with anti-PD-1 synergizes with transfer of tumour-specific Tcm cells, increasing CD8[+] T-cell infiltration of tumours. In addition, Batf3-dependent DCs are crucial for reactivation of circulating CD8[+] T-cell memory, inducing anti-tumour immunity. Knowledge on the generation of optimal memory against tumour antigens is essential for improved cancer immunotherapy.

## Results

**Trm and circulating memory promote anti-tumour response.** To investigate the potential interplay between circulating memory and Trm CD8[+] T cells in anti-tumour immunity we first infected mice with rVACV-OVA by different routes and measured circulating and resident memory at 30 d.p.i. Frequencies of endogenous OVA-specific circulating memory T cells were similar regardless the infection route (Fig. 1a and Supplementary Fig. 1a). Whereas intraperitoneal (i.p.) infection with rVACV-OVA was inefficient for the generation of Trm cells in the skin or the lung, skin scarification (s.s.) in the tail promoted Trm cells in the infection site and in a distant cutaneous site, and intranasal (i.n.) infection induced Trm cells in the lung (Fig. 1b–d and Supplementary Fig. 1b–d).

To address the contribution of circulating and Trm CD8[+] T cells to control tumour growth, mice were infected with rVACV-OVA by s.s. or i.p. and, after generation of resident and/or circulating memory from the endogenous repertoire 30 days later[12], were inoculated intradermally (i.d.) with B16-OVA cells (Fig. 2a). We used the S1P antagonist FTY720 to block egress of T cells from LN into blood[12] and, in this way, limit the contribution of circulating memory T cells to the recall response. FTY720 administration at 30 days following rVACV-OVA i.p. or s.s. infection significantly reduced the presence of blood OVA-specific T circulating memory to the numbers observed in naive mice (Fig. 2b,c). Circulating memory T cells generated by i.p. vaccination with rVACV-OVA were sufficient to delay B16-OVA melanoma growth and this effect was significantly impaired by administration of FTY720 just before tumour inoculation and during tumour growth (Fig. 2d), demonstrating that circulating memory T cells protect against tumour development. However, following s.s. with rVACV-OVA, FTY720 treatment failed to reverse the enhanced tumour rejection (Fig. 2e), suggesting that Trm cells generated by this route are also sufficient for effective anti-tumour immunity. Thus, both circulating memory and Trm CD8[+] T cells are sufficient for anti-tumour immunity, resulting in delayed growth of B16-OVA-derived melanoma.

**Trm improve anti-tumour immunity.** To study the potential contribution of Trm cells to control melanoma growth in the presence of circulating memory T cells, we employed a parabiosis strategy with mice sharing circulating memory, but with only one of the parabionts bearing Trm (Fig. 3a). Accordingly, mice were infected by s.s. with rVACV-OVA 30 days before surgical parabiosis with naive mice and, after allowing a further 30-day period to equilibrate circulating memory T cells (Fig. 3b), parabiont mice were separated, allowed to recover and injected i.d. with B16-OVA cells. When compared with tumour-challenged naive mice, both infected or uninfected parabionts were significantly protected (Fig. 3c,d). However, when compared with uninfected partners that shared circulating memory T cells, detectable tumour onset was delayed in infected parabionts containing Trm cells (Fig. 3c). The delayed tumour incidence in infected parabionts containing Trm cells also resulted in significantly reduced tumour growth (Fig. 3d). Thus, the presence of Trm cells together with circulating memory T cells improves anti-tumour immunity in contrast to an environment containing only circulating memory CD8[+] T cells.

**Plasticity of Tcm to become Trm upon viral challenge.** Next, we investigated the potential plasticity of circulating memory CD8[+] T cells to produce Trm cells. Naive OT-I (CD44[lo]CD62L[hi]) cells are plastic and can become both circulating and skin-resident memory T cells upon transfer to mice subsequently infected in the skin with rVACV-OVA to favour Trm cell differentiation[13,24]. Mice transferred with naive OT-I cells and challenged i.d. with rVACV-OVA were used as source for OT-I Tcm cells 30 d.p.i. (Fig. 4a). Using naive OT-I cells as a positive control, we tested whether transfer of OT-I Tcm cells (CD44[hi] CD62L[hi]) could generate Trm cells in the skin 30 days or 60 days after infection i.d. with rVACV-OVA along with Tcm or naive OT-I transfer (Fig. 4a). Notably, Tcm cells generated Trm cells, defined as OT-I T cells in the skin 30 days or 60 days after viral infection and OT-I Tcm cell transfer, albeit with a lower efficiency than naive T cells (Fig. 4b,c). The majority of skin OT-I Trm cells derived from transferred OT-I Tcm cells showed stable CD69 expression with 50% of them co-expressing CD103 (Fig. 4d), which is consistent with previous results[12,13].

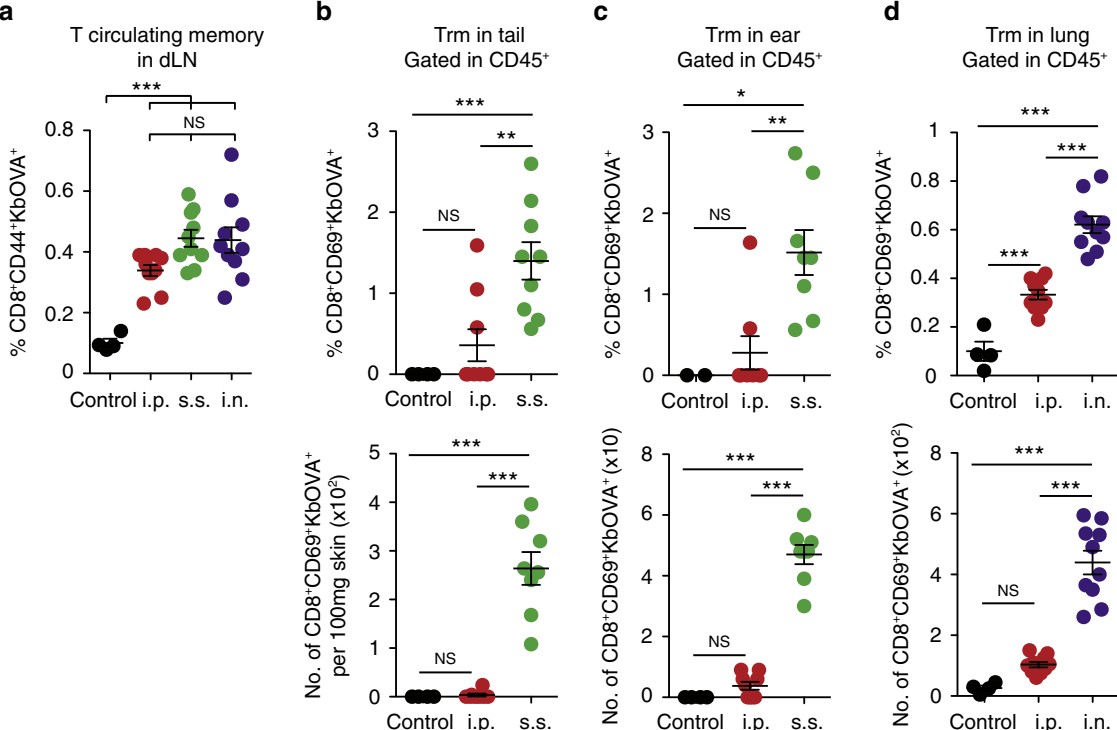

**Figure 1 | Generation of Trm cells after different routes of rVACV-OVA infection.** (**a**) Frequency of endogenous OVA-specific circulating memory CD8+ T cells in the draining LN (dLN) 30 days after i.p. ($5 \times 10^4$ p.f.u.), s.s. ($2 \times 10^6$ p.f.u.) or i.n. ($5 \times 10^4$ p.f.u.) infection with rVACV-OVA. (**b–d**) Frequency (top) and numbers (bottom) of endogenous OVA-specific Trm cells in the tail (**b**) and in the ear (**c**) 30 days after s.s. in the tail, and in the lung (**d**) 30 days after i.n. infection with rVACV-OVA. (**a–d**) Pool of two independent experiments represented as individual data and mean ± s.e.m. ($n = 4–5$ per group). NS, not significant, ***$P < 0.001$; **$P < 0.01$; *$P < 0.05$ by one-way ANOVA, with Bonferroni *post-hoc* test.

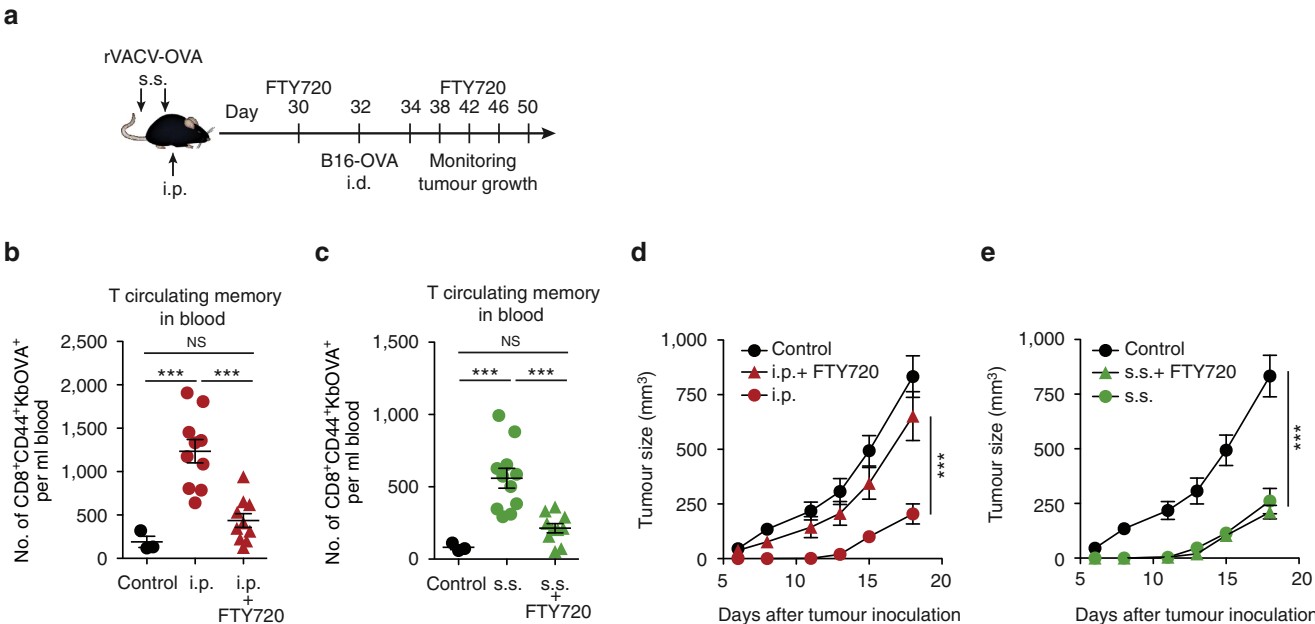

**Figure 2 | Resident and circulating memory CD8+ T cells are sufficient for anti-tumour immunity.** (**a**) Scheme for FTY720 administration. Mice were s.s. or i.p. vaccinated with rVACV-OVA. Starting at day 30, mice were treated i.p. with 50 μg FTY720 every 4 days. At day 32, mice were inoculated (i.d.) with B16-OVA ($10^6$ cells) in the flank. (**b,c**) Absolute numbers of CD8+CD44+KbOVA+ circulating memory T cells in the blood 32 d.p.i. and treated or not with FTY720 at day 30. (**d,e**) B16-OVA growth curve plotted as tumour size ($mm^3$) over time. Simultaneous experiments compared to the same control mice. Pool of two independent experiments represented as individual data and mean ± s.e.m. (**b,c**) and as tumour size mean ± s.e.m. (**d,e**) ($n = 5–6$ per control group and 7–8 per vaccinated group). NS, not significant, ***$P < 0.001$ by one-way ANOVA (**b,c**), and two-way ANOVA (**d,e**) with Bonferroni *post-hoc* test.

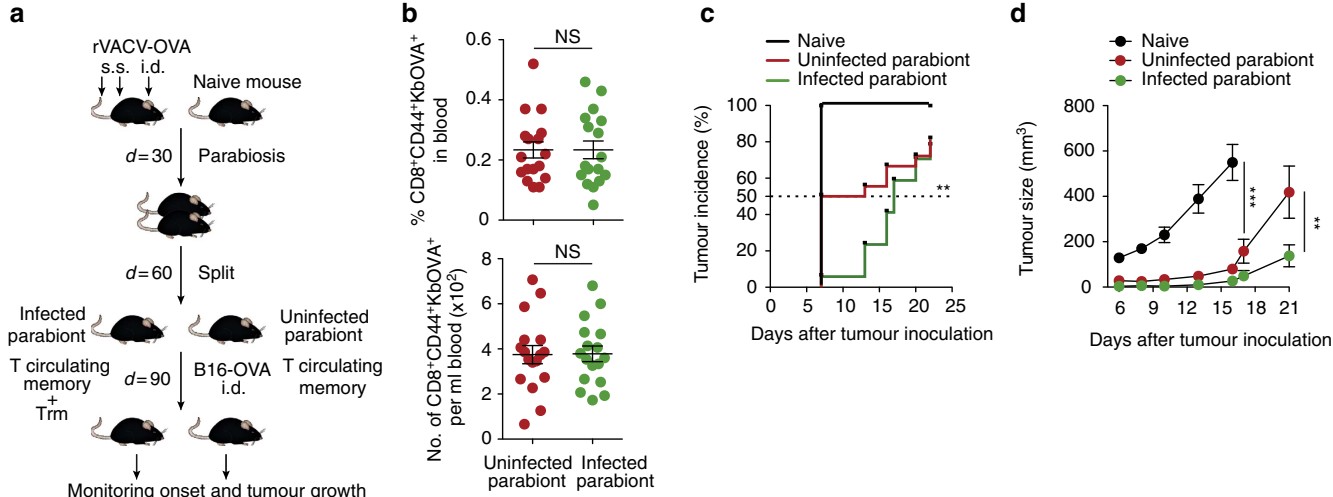

**Figure 3 | Trm cells improve circulating memory-mediated immunity to melanoma.** (**a**) Scheme for parabiosis strategy. Mice were s.s. and i.d. vaccinated with rVACV-OVA. After 30 days, vaccinated mice were each surgically joined with a naive mouse. Thirty days after surgery, parabiotic pairs were separated and allowed to recover for 30 days before i.d. inoculation of B16-OVA ($10^6$ cells) in the flank. (**b**) Frequency (top) and absolute numbers (bottom) of CD8$^+$CD44$^+$KbOVA$^+$ circulating memory T cells in the blood after parabiosis and before tumour inoculation. (**c**) Tumour incidence represented as the frequency of detectable tumours incidence over time. The dashed line indicates 50% of tumour incidence. (**d**) Tumour growth curve plotted as tumour size (mm$^3$) over time. Pool of two independent experiments represented as individual data and mean ± s.e.m. (**b**), as percentages (**c**) and as tumour size mean ± s.e.m. (**d**) ($n = 5$–6 per control group and 8–9 per vaccinated group). NS, not significant, ***$P < 0.001$; **$P < 0.01$ by two-tailed unpaired Student's t-test (**b**), Cox mixed model (**c**) and two-way ANOVA (**d**) with Bonferroni *post-hoc* test.

To functionally demonstrate that OT-I Trm cells derived from Tcm cells following viral infection were truly resident, we tested their ability to migrate via blood or lymph once they were established (Fig. 4e). Mice were transferred with OT-I Tcm cells one day before infection with rVACV-OVA. After 30 d.p.i., surgical parabiosis of infected and naive mice was performed, leaving 30 days to equilibrate circulating memory compartments. Infected and non-infected parabionts were subsequently analysed for the presence of OT-I circulating memory cells in the spleen (Fig. 4f) and Trm cells in the ear (Fig. 4g,h). Only the infected parabionts exhibited OT-I Trm cells in the ear skin, functionally demonstrating that Trm derived from Tcm cells are unable to migrate via blood or lymph.

**Tcm give rise to Trm cells upon tumour inoculation.** To analyse whether Tcm cell plasticity to generate Trm cells is also found in the tumour setting, OT-I Tcm cells were transferred to recipient mice that were subsequently inoculated i.d. with B16-OVA cells or injected subcutaneously (s.c.) with MC38-OVA (Fig. 5a). Following 20 days of B16-OVA tumour development, OT-I cells with a Trm cell-like phenotype, with most OT-I T cells expressing CD69 and 50% of them co-expressing CD103, were found in the tumour mass (Fig. 5b–d). In addition, since Tcm transfer resulted in elimination of MC38-OVA (Fig. 5e), we analysed the presence of Trm cells in the skin that was in the proximity of the rejected tumour. Skin OT-I Trm cells expressing CD69 and CD103 were found 20 days and 45 days after tumour inoculation (Fig. 5f,g and Supplementary Fig. 2a,b). These data support the notion that Tcm retain potential to generate Trm cells upon tumour challenge. Such plasticity also contributes to explain that circulating memory T cells are sufficient for anti-tumour immunity.

**Anti-PD-1 boosts Trm cells in the tumour after Tcm transfer.** Next, we wondered whether anti-PD-1 would synergize with Tcm cell transfer for improved tumour immunotherapy. Indeed, Tcm cells in the tumour draining LN and, particularly, Trm-like cells infiltrating B16-OVA or MC38-OVA grafted tumours showed high PD-1 expression (Fig. 6a,b and Supplementary Fig. 3a,b), suggesting that their function could be enhanced by PD-1 blockade. We thus administered anti-PD-1 antibody concomitantly to Tcm transfer in a tumour therapy setting (Fig. 6c). The combination of Tcm and anti-PD-1 delayed the development of i.d. B16-OVA tumour (Fig. 6d) and s.c. MC38-OVA tumour (Fig. 6e) when compared to the treatment with Tcm cells alone. Notably, tumour-infiltrating lymphocytes with a Trm phenotype were increased in numbers and frequencies within CD45$^+$ cells more than tenfold in average upon anti-PD-1 treatment in both tumour settings (Fig. 6f,g and Supplementary Fig. 3c). In contrast, anti-PD-1 treatment did not affect the numbers or frequencies of OT-I Tcm cells in LNs draining B16-OVA (Supplementary Fig. 3d) or MC38-OVA (Supplementary Fig. 3e) tumours. These data show that anti-PD-1 treatment increases Trm-like tumour cell infiltrate and improves anti-tumour immunity following adoptive immunotherapy with Tcm cells.

**Anti-tumour memory response is impaired in *Batf3*$^{-/-}$ mice.** The generation of Trm cells but not circulating memory T cells by VACV infection is dependent on Batf3-dependent cross-presenting DCs[13]. Consistent with this, we found that generation of endogenous repertoire OVA-specific Trm cells induced by s.s. with rVACV-OVA (Fig. 7a,b), but not circulating memory CD8$^+$ T cells (Fig. 7c), was impaired in *Batf3*$^{-/-}$ mice. We therefore hypothesized that impaired Trm cell generation in *Batf3*$^{-/-}$ mice could lead to a defective anti-tumour response. We found that Batf3 significantly contributed to anti-tumour immunity following s.s. of mice with rVACV-OVA (Fig. 7d). Batf3 was also required for effective anti-tumour response after i.n. infection with rVACV-OVA and subsequent intravenous (i.v.) challenge with B16-OVA cells (Fig. 7e). These results supported our initial hypothesis, as both s.s. and i.n. routes of infection generate Trm cells. However, Batf3 was also required to control tumour growth following routes that do not produce Trm, such as i.p. infection with rVACV-OVA and i.d. challenge with B16-OVA cells (Fig. 7f) or following i.p. infection with

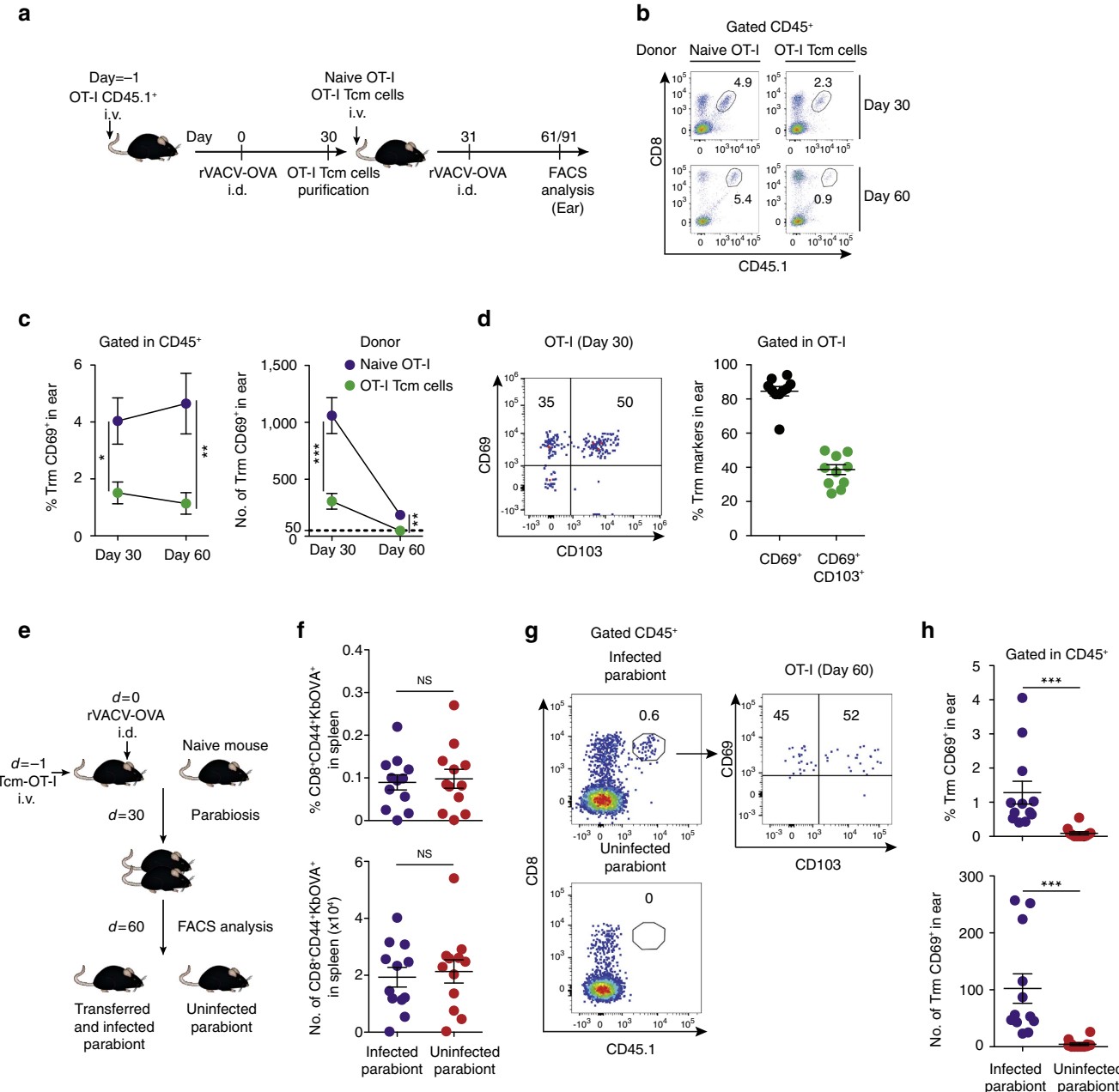

**Figure 4 | Plasticity of central memory T cells to become Trm cells upon viral infection.** (**a**) Scheme for testing Tcm cells plasticity. For generation of Tcm cells, mice were transferred with OT-I CD45.1[+] T cells (1–3 × 10[5] cells) and subsequently i.d. infected with rVACV-OVA (5 × 10[4] p.f.u.) in the ear. After 30 days, Tcm cells were sorted and transferred as indicated. (**b–d**) Mice were transferred with naive OT-I CD45.1[+] T cells or OT-I CD45.1[+] Tcm cells one day before i.d. challenge with rVACV-OVA in the ear. (**b**) Representative FACS dot-plots showing OT-I cells in the CD45[+] population in the ear at the indicated days p.i. (**c**) Frequency within CD45[+] cells (left) and absolute numbers (right) of CD69[+]CD8[+] T cells in the ear at the indicated day p.i. (**d**) Representative FACS dot-plots (left) and frequency (right) of Trm expression markers in OT-I cells 30 d.p.i. (**e**) Scheme showing the parabiosis strategy. Mice were transferred with OT-I CD45.1[+] Tcm cells before i.d. ear infection with the rVACV-OVA. After 30 days, transferred and vaccinated mice were each surgically joined with a naive mouse. Thirty days after surgery, the ears of parabiont pairs were analysed for Trm detection by FACS. (**f**) Frequency (top) and absolute numbers (bottom) of CD8[+]CD44[+]KbOVA[+] circulating memory T cells in the spleen after 30 days of parabiosis. (**g**) Representative FACS dot-plots for OT-I cells identification in CD45[+] population in the ear (left top, infected parabiont; left bottom, uninfected parabiont) with a Trm phenotype (right, infected parabiont). (**h**) Frequency (top) and absolute number (bottom) of CD69[+]CD8[+] T cells in the ear of the indicated parabiont. Pool of two independent experiments represented as mean ± s.e.m. (**c**) (n = 5–6 per group), as individual data and mean ± s.e.m. (**d,f,h**) (n = 5–6 per group). NS, not significant, ***P < 0.001; **P < 0.01; *P < 0.05 by two-tailed unpaired Student's t-test (**c,d,f**) and two-tailed nonparametric Mann–Whitney test (**h**).

rVACV-OVA and i.v. challenge with B16-OVA cells (Fig. 7g). Collectively these results demonstrate that Batf3-dependent DCs contribute to the anti-tumour CD8[+] T-cell memory response independently of their role in the generation of Trm cells.

**Reactivation of Tcm cells is mediated by Batf3-dependent DCs.** We next examined the possible mechanisms underlying the crucial need for Batf3-dependent DCs in anti-tumour memory response. To rule out the possibility that circulating memory

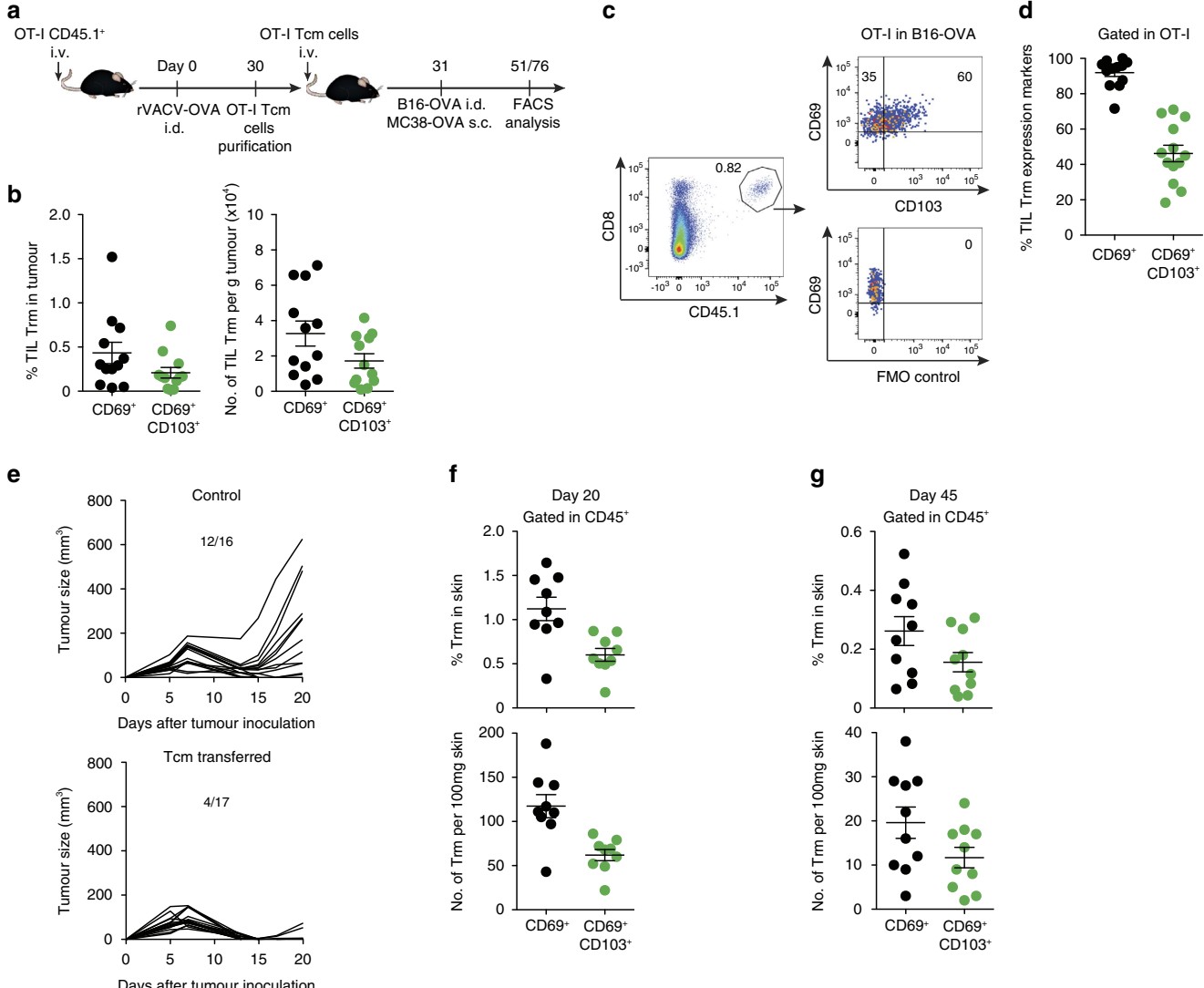

**Figure 5 | Central memory T cells give rise to Trm cells after tumour inoculation.** (**a**) Scheme for testing Tcm cells plasticity. For Tcm cells generation, mice were transferred with OT-I CD45.1$^+$ T cells (1–3 × 10$^5$ cells) and subsequently i.d. infected with rVACV-OVA (5 × 10$^4$ p.f.u.) in the ear. After 30 days, Tcm cells were sorted and used for transfer. Mice were transferred with OT-I CD45.1$^+$ Tcm cells 1 day before i.d. injection with B16-OVA (**b**–**d**) or s.c. injection with MC38-OVA (**e**–**g**) in the flank. (**b**) Frequency (left) and absolute numbers (right) of OT-I cells in B16-OVA-derived cell suspensions with the indicated phenotype in tumour-bearing mice for 23 days. (**c**) Representative FACS dot-plots for OT-I identification (left) and Trm expression markers (right). (**d**) Frequency of Trm expression markers in OT-I cells in B16-OVA-bearing mice. (**e**) MC38-OVA growth curve plotted as individual tumour size (mm$^3$) over time in control mice (top) and in mice treated with Tcm the day before tumour inoculation (bottom). Frequency (top) and absolute numbers (bottom) of OT-I cells in skin-derived cell suspensions with the indicated phenotype, in MC38-OVA-free mice after 20 days (**f**) and 45 days (**g**) of tumour inoculation. Pool of three (**b**,**d**) and two (**e**–**g**) independent experiments represented as individual data and mean ± s.e.m. (**b**,**d**,**f**,**g**) (n = 3–5 per group) and as growth curve of individual tumours (**e**) (n = 5 and 11 per control group and n = 10 and 7 per Tcm transferred group).

CD8$^+$ T cells raised in *Batf3*$^{-/-}$ mice were not functional for anti-tumour immunity, we i.d. infected WT and *Batf3*$^{-/-}$ mice with rVACV-OVA following OT-I cells transfer, and then transferred purified OT-I Tcm cells to WT recipients (Fig. 8a). Following i.d. challenge with B16-OVA cells, Tcm cell transfer was equally effective in delaying tumour growth irrespective of whether they were generated in WT or in *Batf3*$^{-/-}$ mice (Fig. 8b). We thus hypothesized that recipient Batf3-deficient DCs would mediate inefficient reactivation of the transferred Tcm against the tumour. To test this, we generated OT-I Tcm cells in WT (Fig. 8c) or *Batf3*$^{-/-}$ (Fig. 8d) donor mice and transferred them to WT or *Batf3*$^{-/-}$ recipient mice that were i.d. challenged with B16-OVA cells. Independently of the origin of Tcm cells, transfer to *Batf3*$^{-/-}$ recipients resulted in impaired anti-tumour immunity with respect to transfer to WT recipients (Fig. 8c,d).

These data demonstrate that Batf3-dependent DCs are requisite for reactivation of Tcm cells promoting anti-tumour immunity.

## Discussion

Optimal tumour immunotherapy should generate a potent memory CD8$^+$ T cell to prevent local relapse and metastasis. Memory CD8$^+$ T cell can be either circulating or resident in the tissues (Trm) exhibiting different priming and differentiation requirements[3,5,6,13]. Circulating memory and Trm cells also show different effector behaviour[6], but their interplay in tumour immunity remains poorly characterized. In the present study, we demonstrate that circulating memory CD8$^+$ T cells and Trm cells cooperate in anti-tumour immunity, with the circulating memory compartment retaining enough degree of plasticity to

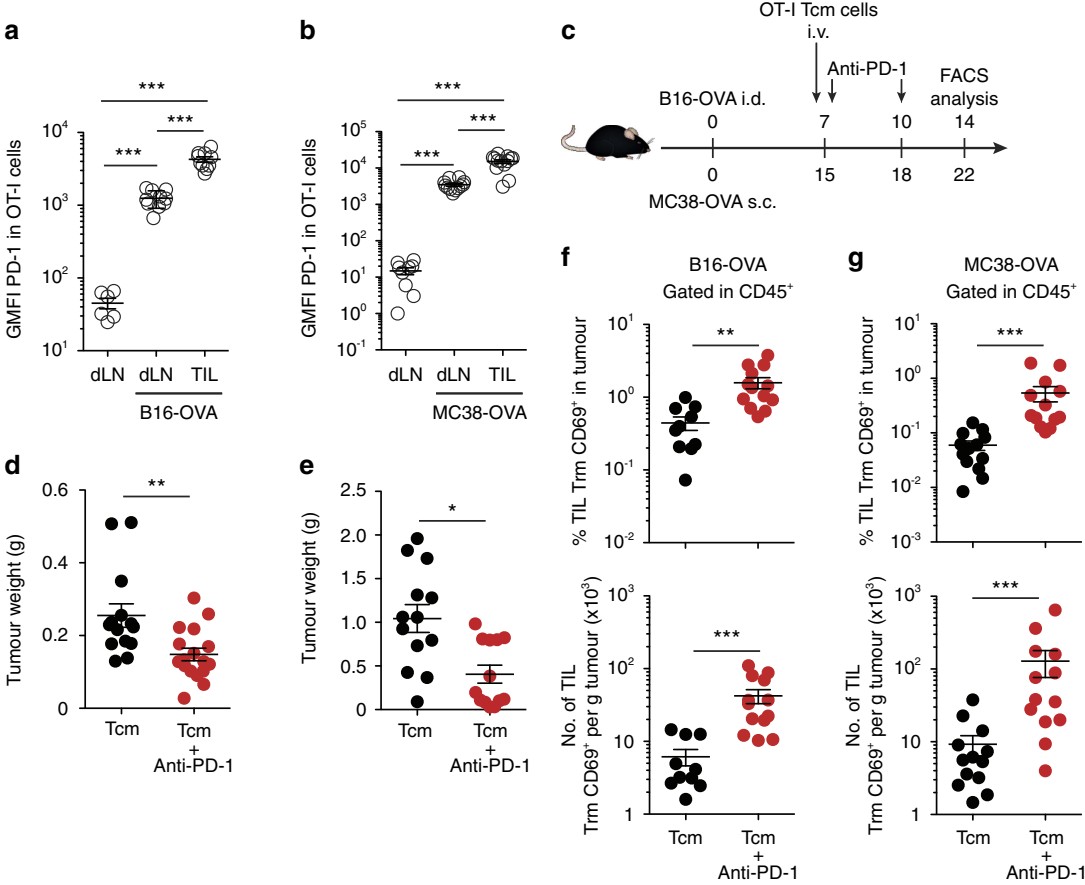

**Figure 6 | Anti-PD1 boosts Trm-like cells in the tumour after Tcm transfer.** (**a,b**) Untreated mice or mice injected i.d. with B16-OVA (**a**) or s.c. with MC38-OVA (**b**) were transferred with OT-I Tcm cells. Expression of PD-1 in OT-I cells was analysed 7 days later in draining LNs (dLN) or tumours. (**c**) Mice were injected with the indicated tumours, transferred i.v. with OT-I CD45.1[+] Tcm cells, and treated with anti-PD-1. (**d,e**) tumour weight at the time of killing following B16-OVA (**d**) or MC38-OVA (**e**) inoculation. (**f,g**) Frequency (top) and numbers (bottom) of OT-I CD69[+] infiltrating B16-OVA (**f**) or MC38-OVA (**g**) tumours in mice treated as indicated (**c**). Pool of four (**d**) and three (**a,b,e–g**) independent experiments represented as individual points and mean ± s.e.m. (**a,b**) (n = 2–3 per control group, 3–4 per tumour group), (**d–g**) (n = 3–5 per group). NS, not significant, ***$P < 0.001$; **$P < 0.01$; *$P < 0.05$ by one-way ANOVA (**a,b**) with Bonferroni *post-hoc* test, two-tailed unpaired Student's *t*-test (**f** top) and two-tailed nonparametric Mann–Whitney test (**d-f** bottom, **g**).

become resident memory cells within the grafted tumour or in the proximal skin following tumour elimination. Notably, anti-PD-1 therapy synergizes to improve anti-tumour immunity following Tcm transfer. Moreover, Batf3-dependent DCs are crucial for reactivation of circulating CD8[+] T-cell memory for anti-tumour immunity.

Trm cells are generated following most viral infections affecting the skin or mucosae[6] and are broadly distributed in tissues[7]. Trm cells respond faster to stimulation than circulating memory T cells, and they are equipped with a more potent effector response than their circulating counterparts[11]. Moreover, Trm cells provide superior protection upon viral reinfection in skin or mucosae and are sufficient to control viral reinfection in the presence of FTY720, which prevents the contribution of circulating memory T cells[12]. Our experiments using FTY720 showed that Trm were potentially sufficient for anti-tumour immunity. However, we also found that circulating memory CD8[+] T cells mediate effective anti-tumour immunity, in agreement with previous results[25]. Using a parabiosis strategy, we demonstrate that melanoma tumour growth is slower in mice containing both Trm and circulating memory T cells than in parabionts containing only circulating memory T cells, thus supporting the notion that resident and circulating memory subsets cooperate for enhanced anti-tumour immunity. The

differential kinetics in the effector response of resident and circulating memory subsets has been shown in other experimental settings, such as cutaneous hypersensitivity, where Trm mediate rapid responses, whereas circulating memory T cells mediate delayed hypersensitivity[26]. Conceivably upon viral rechallenge, Trm cells trigger a local alarm state that promotes immunity[9,10,27]. Our data showing delayed tumour incidence and growth in the presence of Trm suggest that these cells could be particularly efficient in preventing metastasis.

The plasticity of Tcm cells to generate Trm cells had not been previously addressed. Tcm and Trm cells have a clonal origin, sharing TCR repertoires[26], but display distinct priming requirements in the LN[13]. Optimal generation of Trm cells following rVACV infection requires unique priming signals provided by Batf3-dependent DCs that favour T-bet expression and LN retention[13]. Subsequently, Trm cells migrate to the tissue, where the local microenvironment conditions Trm cells differentiation[6,7,28]. Tcm cells exhibit more stem-like and proliferative capacities after re-exposure to antigen, whereas Trm cells have a higher effector capacity and trigger a local alarm state[8–10]. We found that adoptive transfer of Tcm cells generates skin Trm cells upon rVACV infection. Moreover, following tumour challenge, transferred Tcm convert to cells with Trm phenotype within the tumour or in the proximal skin following

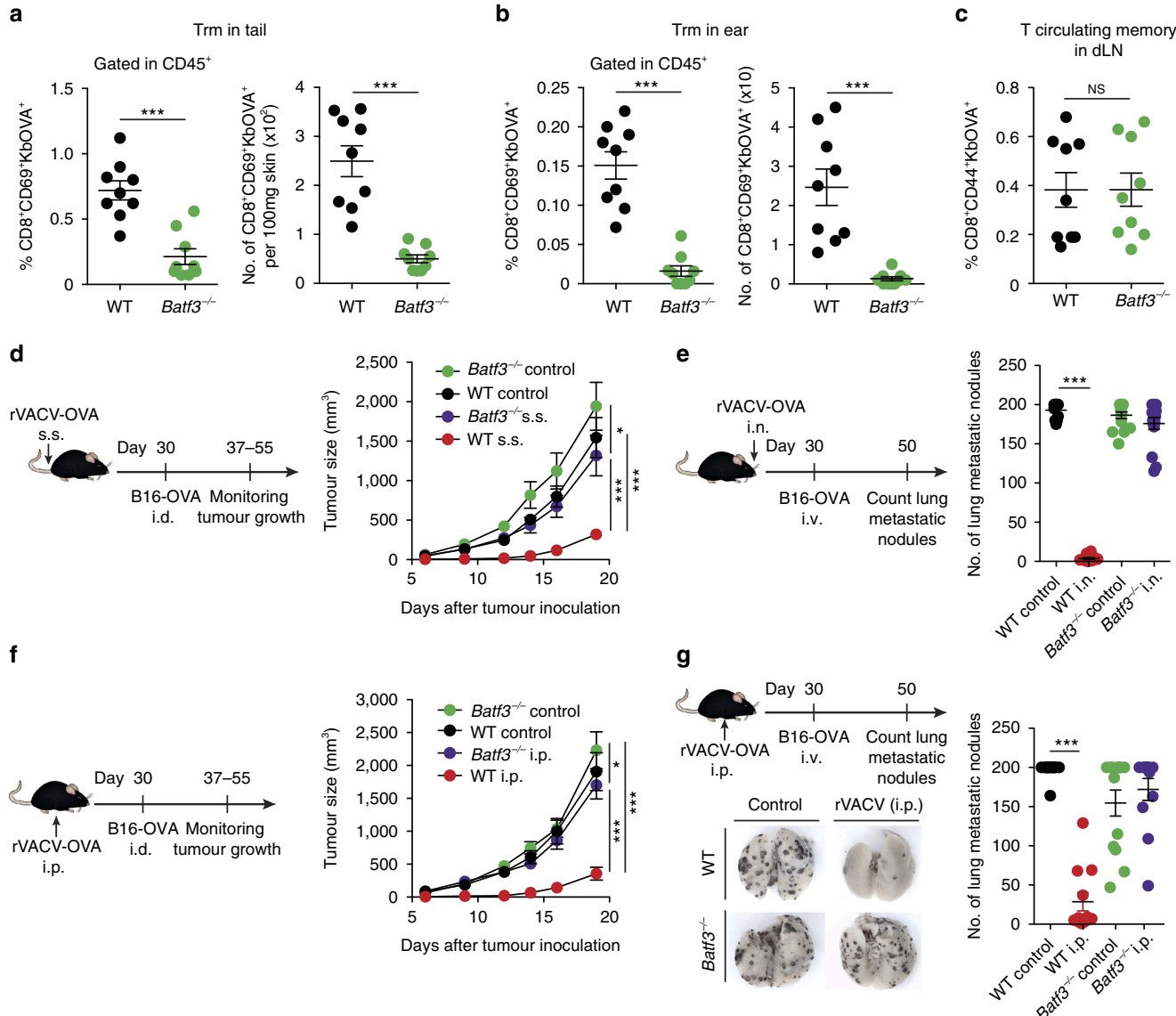

**Figure 7 | Anti-tumour memory response is impaired in Batf3−/− mice.** (**a–c**) Frequency (left) and absolute numbers (right) of endogenous OVA-specific Trm cells in the tail (**a**) and in the ear (**b**), and frequency of Kb-OVA$^+$ circulating memory T cells in the draining LNs (dLN) (**c**) 30 days after s.s. infection with rVACV-OVA in the tail of WT and Batf3−/− mice. (**d–g**) WT and Batf3−/− mice were infected with rVACV-OVA by s.s. in the tail (**d**) i.n. (**e**) or i.p. (**f,g**). After 30 days, mice were i.d. inoculated with B16-OVA in the flank (**d,f**) and B16-OVA growth curve plotted as tumour size (mm$^3$) over time (right). Alternatively, B16-OVA was injected i.v. ($3 \times 10^5$ cells) 30 days later (**e,g**) and number of lung B16-OVA nodules after 20 days since intravenous tumour challenge (right). (**g**) Representative images of lung B16-OVA metastatic nodules (bottom left). Pool of two independent experiments represented as individual data and mean ± s.e.m. (**a–c,e,g**) ($n = 5$–7 per group) and as tumour size mean ± s.e.m. (**d,f**) ($n = 6$–8 per group). NS, not significant, ***$P < 0.001$; *$P < 0.05$ by two-tailed unpaired Student's t-test (**a–c**), two-way ANOVA (**d,f**) and one-way ANOVA (**e,g**) with Bonferroni post-hoc test.

tumour rejection. The capacity to generate Trm cells subsets upon reinfection or tumour challenge supports at least partial stemness properties of Tcm cells, which do not convert into Trm under steady-state conditions in the absence of viral infection or tumour implantation[12].

DCs are essential for anti-tumour immunity by adoptively transferred T cells, since transfer of preactivated OT-I cells to DC-depleted mice leads to impaired protection against tumour cells expressing OVA[15]. In the tumour context, recent studies show that Batf3-dependent CD103$^+$ DCs play a crucial role in anti-tumour immunity[15–18,29]. Batf3-dependent DCs cross-present tumour antigens and generate a baseline anti-tumour response that can be potentiated by immunostimulating

antibodies in synergy with expansion and activation of this DC subset using Flt3L and poly I:C (refs 17,18). Batf3-dependent DCs are also major producers of IL-12, not only in infectious settings[30–32], but also in the context of tumours, where IL-12 contributes to CD8 effector function[29]. In addition, Batf3-dependent DCs are required for the recruitment of naive CD3$^+$ T cells to the tumour site[16] in a spontaneous melanoma model. Here we show that Batf3-dependent DCs are needed for effective reactivation of Tcm cells to promote anti-tumour immunity, which further supports the crucial role of this DC subset in tumour immunology and immunotherapy, not only for the primary response, but also for the memory response. Our data in the tumour context concur with previous

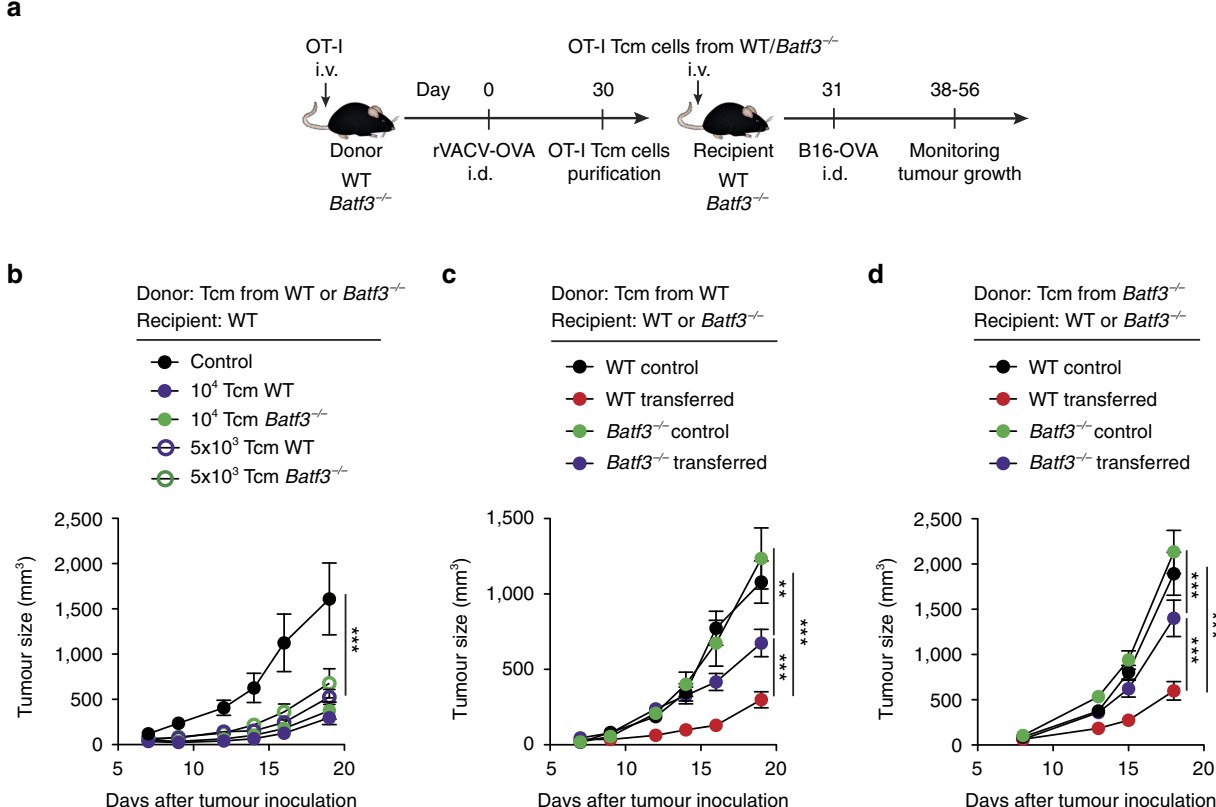

**Figure 8 | Reactivation of anti-tumour Tcm cells is mediated by Batf3-dependent DCs.** (**a**) For Tcm cell generation, WT and/or $Batf3^{-/-}$ mice were transferred with OT-I CD45.1[+] T cells 1 day before i.d. infection with rVACV-OVA in the ear. After 30 days, OT-I CD45.1[+] Tcm cells from donor mice were sorted and transferred to WT and/or $Batf3^{-/-}$ recipient mice. One day after transfer, mice were inoculated with B16-OVA cells (i.d.) in the flank. (**b–d**) B16-OVA growth curve plotted as tumour size ($mm^3$) over time in the indicated recipient mice transferred with Tcm cells from the indicated donor mice. Pool of two independent experiments represented as tumour size mean ± s.e.m. (**b–d**) ($n = 6$–8 per group). NS, not significant, ***$P < 0.001$; **$P < 0.01$ by two-way ANOVA (**b–d**) with Bonferroni *post-hoc* test.

data on the role of Batf3-dependent DCs in reactivation of memory CD8[+] T-cell recall response upon infection with Listeria monocytogenes, vesicular stomatitis virus or vaccinia virus[33]. We cannot rule out, however, that the role of Batf3-dependent DCs in generation of Trm[13] is also important in the context of tumours or that additional DC subsets may activate Trm[34].

Our results suggest that CD8-mediated anti-tumour immunity arises from the interplay between resident and circulating memory CD8[+] T cells. Circulating memory cells show enough plasticity to generate Trm cells upon tumour challenge and they both express PD-1, similarly to CD103[+] T cells infiltrating high-grade serous epithelial ovarian cancer[35]. Thus, anti-PD-1 therapy synergizes with transfer of Tcm cells for improved anti-tumour immunity, increasing the infiltration of Trm-like cells expressing PD-1 within the tumour. These results concur with previous data showing that anti-PD-1 expands intratumoural memory T cells in patients[36]. Within samples of melanoma, immune checkpoints are particularly enriched within T cells with phenotype and genomic features of Trm cells, suggesting that the Trm subset of TILs may be the major target of immune checkpoint blockade[37]. Moreover, other immunostimulatory antibodies, such as anti-CD137, could also enhance the resident memory response[38]. In conclusion, our results support the notion that anti-tumour vaccination strategies should aim at the generation of both circulating and resident memory CD8[+] T-cell subsets[22,23,39], which could synergize with checkpoint antibody therapy for improved cancer immunotherapy.

## Methods

**Mice.** Mice were bred and housed at the CNIC animal facility in specific pathogen-free conditions. $Batf3^{-/-}$ mice on the C57BL/6 background were kindly provided by Dr K.M. Murphy (Washington University, St Louis, MO, USA)[14]. OT-I transgenic mice (C57BL/6-Tg (TcraTcrb)1100Mjb/J) were mated with B6-SJL (Ptprca Pepcb/BoyJ) mice expressing the CD45.1 allele, both from The Jackson Laboratory (Bar Harbor, ME, USA). We used 7- to 10-week-old animals (males or females) for all experiments. Experiments were repeated 2–3 times to reach statistical significance. No blinding or randomization strategy was used and no animal was excluded from analysis. The local ethics committee approved all animal studies. All animal procedures conformed to EU Directive 2010/63EU and Recommendation 2007/526/EC regarding the protection of animals used for experimental and other scientific purposes, enforced in Spanish law under Real Decreto 1201/2005. Mice were allocated randomly in the different experimental procedures.

**Viral infection and tumour challenge.** Recombinant vaccinia virus expressing full-length ovalbumin (OVA) protein (rVACV-OVA) was a gift from J.W. Yewdell and J.R. Bennink (NIH, Bethesda, MD, USA) and was kindly provided by M. del Val (CBMSO, Madrid, Spain). Growth of viral stocks and titration was performed as described in CV1 cells[40]. Mice were infected with rVACV-OVA by the following routes: s.s. at the base of the tail (1–2 × $10^6$ p.f.u.) or in the back ($10^6$ p.f.u.), i.d. in the ear pinnae (5 × $10^4$ or $10^6$ p.f.u.), i.n. (5 × $10^4$ p.f.u.) or i.p. (5 × $10^4$ or 1–2 × $10^6$ p.f.u.). T cells were considered memory cells 30 d.p.i (ref. 12). Mice were inoculated with the OVA-expressing B16 melanoma cell line (B16-OVA)[17] i.d. in the flank ($10^6$ cells) or OVA-expressing MC38 tumour cell line (MC38-OVA)[17] s.c. in the flank (2 × $10^6$ cells), and tumour growth was monitored for 20–30 days. Tumour volumes were calculated using the following formula: $V = D \times d^2/2$, where $V$ is volume ($mm^3$), $D$ is larger diameter (mm) and $d$ is smaller diameter (mm). Alternatively, mice were injected i.v. with B16-OVA (3 × $10^5$ cells) and killed 20 days later. Lungs were fixed in Fekete's solution and tumours were counted. FTY720 (Cayman Chemical) was administered i.p. at a dose of 2.5 mg kg[−1] in aqueous solution every 4 days. All cell lines used were tested for mycoplasma routinely.

**Generation and analysis of OT-I Tcm and Trm cells.** To generate OT-I Tcm cells, mice were transferred i.v. with $1–3 \times 10^5$ naive OT-I cells 1 day before i.d. ear infection with rVACV-OVA ($5 \times 10^4$ p.f.u.). Tcm cells were FACS-sorted (Sy3200, Sony) from spleens and draining LNs 30 d.p.i. Mice were transferred with $10^4$ naive OT-I or $1–2 \times 10^4$ OT-I Tcm cells 1 day before virus challenge, with $2 \times 10^4$ OT-I Tcm cells 1 day before B16-OVA inoculation and $3 \times 10^3$ or $2 \times 10^4$ OT-I Tcm cells 1 day before MC38-OVA inoculation. For virus challenge, mice were infected with rVACV-OVA ($5 \times 10^4$ p.f.u.) i.d. in the ear, and the memory response was analysed at 30 days or 60 d.p.i. Tumour inoculation (B16-OVA cells, i.d.; MC38-OVA cells, s.c.) was performed as indicated above. For analysis of tumour-infiltrating lymphocytes, mice were killed 20–23 days after tumour inoculation. For analysis of skin-infiltrating lymphocytes, mice were killed 20 or 45 days after tumour inoculation. For anti-PD-1 antibody treatment, mice were transferred with $1–2 \times 10^4$ OT-I Tcm cells when the tumour reached $100–150\,\text{mm}^2$ (day 7, for B16-OVA; day 15–18, for MC38-OVA), and inoculated i.p. the same day of the Tcm transfer and 3 days later, with $100\,\mu\text{g}$ of anti-PD-1 antibody (RMP1-14, BioXcell). For FACS analysis mice were killed at day 7 after Tcm transference.

**Parabiosis.** Parabiosis was performed as described[12]. For tumour inoculation, wild-type (WT) mice were infected with rVACV-OVA ($10^6$ p.f.u.) by s.s. in the tail and in the back and i.d. in the ear 30 days before surgical joining with naive mouse to create parabionts. After 30 days, mice were separated and kept for recovery after surgery for 30 days. Finally, both parabiont mice were i.d. challenged with B16-OVA cells as indicated. To analyse the migratory capacity of Trm cells that derive from Tcm cells, WT mice were transferred with $2 \times 10^4$ OT-I Tcm cells i.v. one day before i.d. infection with rVACV-OVA ($5 \times 10^4$ p.f.u.) in the ear. After 30 days, infected mice were joined with naive mice and kept in parabiosis for 30 days. Finally, ears and spleens of both parabionts were analysed by FACS.

**Flow cytometry.** Allophycocyanin-labelled dextramers specific for OVA H-2Kb (257-SIINFEKL-264) were purchased from Immudex (Copenhagen, Denmark). Samples for flow cytometry were stained with the appropriate antibody cocktails in ice-cold PBS supplemented with 2 mM EDTA and 1% FBS. Anti-mouse CD45 (clone 30F11), CD8α (clone 53-6.7), CD103 (clone 2E7), CD44 (clone IM7), CD45.1 (clone A20) and CD45.2 (clone 104) antibodies were obtained from eBioscience. Anti-mouse CD62L (clone MEL-14) and CD69 (clone H1.2F3) antibodies were obtained from BD Biosciences. Anti-mouse CD279 (PD-1, clone 29F.1A12) was obtained from Biolegend. Events were acquired using an LSRFortessa SORP (Becton Dickinson) flow cytometer or Spectral Cell Analyzer SP6800 (Sony) and data were analysed using FlowJo V10 software (Tree Star).

**Statistical analysis.** Statistical analysis was performed using Prism v6 (GraphPad Software Inc., La Jolla, CA, USA). We estimated a priori that minimal informative differences are 1 SD, and estimated sample sizes using software 'Gpower 1.3', taking into account the following considerations. (a) The differences to be detected, 1 SD comparing each group of mice. Ratio (effect size) differs to detect/SD = 1. (b) Taking into account the above data, the number of animals in total for a contrast $t$-test of mean differences between two independent groups with two tails is 17. Power = 0.8 and significance level = 0.05. Variance equality among groups was determined using F-test. Statistical significance for comparison between two groups of samples showing a normal distribution (Shapiro–Wilk test for normality) was determined using the unpaired two-tailed Student's $t$-test. For comparison between two groups with a no normal distribution, two-tailed Mann–Whitney nonparametric test was used. For comparison of more than two groups, one-way or two-way ANOVA with Bonferroni post-hoc test was used. For Fig. 3c, Cox mixed model was applied when comparing tumour incidence of two groups, keeping the information about the experimental groups. We used the coxme function of the R package with the same name, to the cohort between day 0 and day 17, where two parabiont groups reach 50% of incidence. A $P$ value < 0.05 was considered significant.

**Data availability.** The authors declare that the data supporting the findings of this study are available within the article and its Supplementary Information files, or available from the authors on request.

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

## Acknowledgements

We are grateful to N. Anandasabapathy, J. Pardo and members of the D.S. lab for discussions and critical reading of the manuscript. We also thank R.A. Mota for the contribution to the development of animal models. We thank the CNIC facilities, personnel and to K. McCreath for editorial assistance. We are indebted to all the scientists who have shared reagents with us, as indicated in Methods. M.E. is the recipient of a CNIC International PhD Programme fellowship 'La Caixa'-Severo Ochoa, 2013 Call (OSLC-CNIC-2013-04). S.I. is funded by grant SAF2015-74561-JIN. I.M. is supported by Asociación Española contra el Cáncer and Fundación BBVA. A.H. is funded by the Spanish Ministry of Economy, Industry and Competitiveness (MEIC) and European Fund for Regional Development (FEDER) (SAF2015-65607-R). D.S. lab is funded by the MEIC and FEDER (SAF-2013-42920-R and SAF-2016-79040-R), and the Fondation ACTERIA. D.S. and I.M. lab are funded by the European Commission (635122-PRO-CROP H2020). D.S. and A.H. lab are funded by the CNIC. The CNIC is supported by the MEIC and the Pro CNIC Foundation, and is a Severo Ochoa Center of Excellence (SEV-2015-0505).

## Author Contributions

M.E. and D.S. conceived and designed the project. M.E., S.I., J.A.Q. and A.H. developed methodology. M.E., E.P. and S.I. acquired the data. M.E., S.I. and D.S. analysed and interpreted the data. M.E. and D.S. wrote and reviewed the manuscript. F.J.C., S.M.-C., E.M.-P., I.M. and M.Est. contributed with administrative, technical and material support. All the authors discussed the results and the manuscript.

## Additional information

**Competing interests:** I.M. is a consultant for BMS, Merck-Serono, Roche-Genentech, AstraZeneca, Heliox, Boehringer-Ingelheim, Alligator and Bioncotech. The remaining authors declare no competing financial interests.

