## [Peer Review File · Nature Communications]

Reviewers' comments:

Reviewer #1 (Remarks to the Author):

In the submitted manuscript by M. Enamorado et al., the authors studied the effect of different vaccination strategies on the generation of resident memory T cells (Trm) and circulating memory T cells. They demonstrated that central memory CD8+ T cells differentiated to cells with a Trm phenotype following viral infection or tumor challenge. Furthermore, reactivation of circulating memory anti-tumor response relies on Batf3-dependent dendritic cells. In addition, the presence of Trm cells improved therapeutic efficacy in melanoma model.

The results are interesting, however some concerns should be addressed.

Major points:

1. It should be clarified which markers were used to identify Trm. Dot plots showing identification of Trm should be presented in Fig. 1.
2. The frequency of Trm infiltrating tumors should be measured.
3. The mechanism of Trm induction through Batf3-dependent DCs should be investigated and discussed.
4. It would be interesting to detect the generation of Trm in another tumor model.
5. How the beneficial effect of PD-1 antibodies injected together with Tcm could be explained?
6. Fig. 4d and 7C are nonvisible (black) and should be replaced.

Reviewer #2 (Remarks to the Author):

In this manuscript, the authors report their studies of interactions between circulating memory CD8+ T cells (Tcm) and resident memory CD8+ T cells (Trm) in the setting of anti-viral or anti-tumor immunity. The authors use mouse models in which various routes of vaccination are used to preferably generate Tcm or Trm immunity or to allow interactions of the two cell populations. The data unambiguously demonstrate that while either Tcm or Trm are sufficient to generate anti-tumor immunity, cooperation between them (with the active participation of Batf3+CD103+ DC) is necessary for efficacy of anti-tumor immune responses. The authors used clever strategies such as parabiosis or blocking of Tcm migration using FTY720 to illustrate collaboration between Tcm and Tcr cells. The demonstration of Tcm plasticity to generate Trm following the viral or tumor challenge is a novel, albeit not surprising, finding: after the tumor challenge, adoptively transferred Tcm converted to TILs with the Trm phenotype.

The manuscript is clear, concise and very well written. The experiments were expertly designed and expertly performed by the highly experienced group of investigators. The manuscript is conceptually strong, and it experimentally confirms the T-cell plasticity and collaboration idea that most immunologists have suspected is critical for T-cell immunity but have not had proof for it. The proof is convincingly presented here. This excellent contribution deserves to be published without revisions.

Point-by-point response to the referees

We thank the reviewers for the constructive criticisms and the positive evaluation of our work. We have addressed their concerns in this revised version, as summarized in the point-by-point reply below. Changes to the main text have been marked in the revised manuscript.

Reviewer #1 (Remarks to the Author):

In the submitted manuscript by M. Enamorado et al., the authors studied the effect of different vaccination strategies on the generation of resident memory T cells (Trm) and circulating memory T cells. They demonstrated that central memory CD8⁺ T cells differentiated to cells with a Trm phenotype following viral infection or tumor challenge. Furthermore, reactivation of circulating memory anti-tumor response relies on Batf3-dependent dendritic cells. In addition, the presence of Trm cells improved therapeutic efficacy in melanoma model.

The results are interesting; however, some concerns should be addressed.

Major points:

1. It should be clarified which markers were used to identify Trm. Dot plots showing identification of Trm should be presented in Fig. 1.

We stained skin or lung CD45⁺CD8⁺ cells 30 d.p.i. using dextramers for the detection of endogenous H-2K^b/OVA₂₅₇₋₂₆₄ (SIINFEKL)-specific T cells induced by rVACV-OVA infection along with the phenotypic marker CD69 (**Fig. 1b-d**), which is stably expressed by Trm. As requested, we have included dot plots depicting gating strategy for identification of circulating memory (**new Supplementary Fig. 1a**) and Trm (**new Supplementary Fig. 1b-d**).

To demonstrate the plasticity of transferred Tcm to generate Trm upon viral infection with rVACV-OVA, we tracked transferred transgenic OT-I Tcm cells 30d or 60d post infection. The Trm-specific phenotypic markers used were CD69 and CD103 (**Fig. 4c-d**). Notably, to further demonstrate functionally that they were bona fide Trm, we have now performed new experiments using parabiosis, in which we demonstrate that Tcm-derived Trm are unable to migrate via blood or lymph once they are established (**new Fig. 4e-h**).

For detection of Trm that derive from Tcm after tumor inoculation, we tracked transgenic OT-I cells expressing CD69 and CD103 within B16-OVA tumors after 23 d of tumor inoculation (**Fig. 5b-d**). We have now performed new experiments in an independent tumor model (MC38-OVA) that is eliminated by the transfer of Tcm and, notably, we found CD69⁺ CD103⁺ CD8⁺ resident T cells in the flank skin 20 d or 45 d after tumor challenge (**new Fig. 5e-g and Supplementary Fig. 2**)

2. The frequency of Trm infiltrating tumors should be measured.

In addition to the numbers, we now show the frequencies of Trm infiltrating tumors as requested. Following Trm generation after Tcm transfer, we show the Trm frequencies in the tumor (**new Fig. 5b, left panel**) and a representative FACS dot-plot showing how CD45.1 OT-I tumor infiltrating lymphocytes express CD69 and CD103 (**new Fig. 5c**).

We also show numbers and frequencies of tumor infiltrating Trm following Tcm transfer and anti-PD1 treatment or not in two different tumor models (**new Fig. 6f and 6g**).

3. The mechanism of Trm induction through Batf3-dependent DCs should be investigated and discussed.

This was the object of another work that was published by our group just before submitting this manuscript and was already quoted and discussed in the previous version (Reference 13, Iborra et al., 2016; DOI: [10.1016/j.immuni.2016.08.019](https://doi.org/10.1016/j.immuni.2016.08.019)). Batf3-dependent DCs induce Trm generation by providing unique priming signals that favor T-bet induction and retention of naive T cells in the LN. As suggested, we have further discussed the mechanisms of Trm induction by Batf3-dependent DCs (page 11, line 19ff).

4. It would be interesting to detect the generation of Trm in another tumor model.

We have analyzed the generation of Trm following Tcm transfer in an additional tumor model (MC38-OVA, **new Fig. 5e-g and Supplementary Fig. 2**). This new set of experiments has added further insights into the generation of Trm in the tumor context, since OT-I Tcm cells transfer was sufficient to eliminate the tumor in most cases (**new Fig. 5e**) and CD69⁺ CD103⁺ transferred OT-I were found in the flank skin next to the rejected tumor at 20 and 45 days after initial tumor inoculation (**new Fig. 5f-g**), further supporting that these are bona fide Trm.

5. How the beneficial effect of PD-1 antibodies injected together with Tcm could be explained?

Our new data show that B16-OVA or MC38-OVA tumors induce the expression of PD-1 in the transferred Tcm, both circulating in dLN and infiltrating the tumor (**new Fig. 6a-b and new Supplementary Fig. 3a-b**). We have found that anti-PD-1 treatment results in a 10-fold increase in infiltration of OT-I Trm cells infiltrating the tumor (**new Fig. 6f and 6g**) while not affecting OT-I numbers in the dLN (**new supplementary Fig. 3d-e**). These results explain the therapeutic effect of anti-PD-1 following OT-I Tcm cells transfer both in B16-OVA (**new Fig. 6d**) and MC38-OVA (**new Fig. 6e**) tumors.

6. Fig. 4d and 7C are nonvisible (black) and should be replaced.

FACS panels (Fig. 4d and 7c in former manuscript, **Fig. 4d and Supplementary Fig. 3c** in revised manuscript) have been replaced, enlarged and now show large dot-plots to improve their visibility.

Reviewer #2 (Remarks to the Author):

In this manuscript, the authors report their studies of interactions between circulating memory CD8⁺ T cells (Tcm) and resident memory CD8⁺ T cells (Trm) in the setting of anti-viral or anti-tumor immunity. The authors use mouse models in which various routes of vaccination are used to preferably generate Tcm or Trm immunity or to allow interactions of the two cell populations. The data unambiguously demonstrate that while either Tcm or Trm are sufficient to generate anti-tumor immunity, cooperation between them (with the active participation of Batf3+CD103⁺ DC) is necessary for efficacy of anti-tumor immune responses. The authors used clever strategies such as parabiosis or blocking of Tcm migration using FTY720 to illustrate collaboration between Tcm and

Tcr cells. The demonstration of of Tcm plasticity to generate Trm following the viral or tumor challenge is a novel, albeit not surprising, finding: after the tumor challenge, adoptively transferred Tcm converted to TILs with the Trm phenotype.

The manuscript is clear, concise and very well written. The experiments were expertly designed and expertly performed by the highly experienced group of investigators. The manuscript is conceptually strong, and it experimentally confirms the T-cell plasticity and collaboration idea that most immunologists have suspected is critical for T-cell immunity but have not had proof for it. The proof is convincingly presented here. This excellent contribution deserves to be published without revisions.

We want to thank the reviewer for the positive evaluation of our work.

REVIEWERS' COMMENTS:

Reviewer #1 (Remarks to the Author):

The authors successfully addressed all my concerns.

Point-by-point response to the referee

Reviewer #1 (Remarks to the Author):

The authors successfully addressed all my concerns.

We want to thank the reviewer for the positive evaluation of our work.